# One health genomic surveillance and response to a university-based outbreak of the SARS-CoV-2 Delta AY.25 lineage, Arizona, 2021

Hayley D. Yaglom[1‡]*, Matthew Maurer[2‡], Brooke Collins[2], Jacob Hojnacki[2], Juan Monroy-Nieto[1], Jolene R. Bowers[1], Samuel Packard[2], Daryn E. Erickson[1,3,4], Zachary A. Barrand[1,3,4], Kyle M. Simmons[1,3,4], Breezy N. Brock[1,3,4], Efrem S. Lim[5], Sandra Smith[6], Crystal M. Hepp[1,3,4], David M. Engelthaler[1]

1 Translational Genomics Research Institute, Flagstaff, Arizona, United States of America, 2 Coconino County Health and Human Services, Flagstaff, Arizona, United States of America, 3 School of Informatics, Computing, and Cyber Systems, Northern Arizona University, Flagstaff, Arizona, United States of America, 4 Pathogen and Microbiome Institute, Northern Arizona University, Flagstaff, Arizona, United States of America, 5 Arizona State University, Tempe, Arizona, United States of America, 6 Campus Health Services, Northern Arizona University, Flagstaff, Arizona, United States of America

‡ HDY and MM contributed equally to this work and co-first authors.
* hyaglom@tgen.org

**Data Availability Statement:** Genomic sequencing data are available through GISAID, a public repository (https://www.gisaid.org/). Relevant data

## Abstract

Genomic surveillance and wastewater tracking strategies were used to strengthen the public health response to an outbreak of the SARS-CoV-2 Delta AY.25 lineage associated with a university campus in Arizona. Epidemiologic and clinical data routinely gathered through contact tracing were matched to SARS-CoV-2 genomes belonging to an outbreak of AY.25 identified through ongoing phylogenomic analyses. Continued phylogenetic analyses were conducted to further describe the AY.25 outbreak. Wastewater collected twice weekly from sites across campus was tested for SARS-CoV-2 by RT-qPCR, and subsequently sequenced to identify variants. The AY.25 outbreak was defined by a single mutation (C18804T) and comprised 379 genomes from SARS-CoV-2 positive cases associated with the university and community. Several undergraduate student gatherings and congregate living settings on campus likely contributed to the rapid spread of COVID-19 across the university with secondary transmission into the community. The clade defining mutation was also found in wastewater samples collected from around student dormitories a week before the semester began, and 9 days before cases were identified. Genomic, epidemiologic, and wastewater surveillance provided evidence that an AY.25 clone was likely imported into the university setting just prior to the onset of the Fall 2021 semester, rapidly spread through a subset of the student population, and then subsequent spillover occurred in the surrounding community. The university and local public health department worked closely together to facilitate timely reporting of cases, identification of close contacts, and other necessary response and mitigation strategies. The emergence of new SARS-CoV-2 variants and potential threat of other infectious disease outbreaks on university campuses presents an

are within the paper and its Supporting Information files.

**Funding:** The NARBHA Institute (#AOTHNARBHA3640C) and the Arizona Department of Health Services (#CTR049427) provided financial support for this work, mainly the genomic sequencing efforts." These agencies had no role in study design, data collection and analysis, decision to publish, or preparation of the manuscript. No authors received a salary from any of the funders.

**Competing interests:** The authors have declared that no competing interests exist.

opportunity for future comprehensive One Health genomic data driven, targeted interventions.

## Introduction

Infectious disease threats faced by university campuses are well-documented, with outbreaks of influenza, meningococcal disease, and other respiratory illnesses having been reported in these settings over the past decade [1–4]. COVID-19 has now joined this list, rapidly unfolding across universities throughout the United States (US) and globally [5–9]. Increasing case counts of COVID-19 and large-scale outbreaks across universities and college campuses, especially after the first year of the pandemic [10], resulted in shifts to online or hybrid learning and the need for rapid implementation of routine testing as well as isolation/quarantine programs [11–13]. The close contact of students in dormitories, dining areas, and lecture halls, combined with common social-behavioral activities puts this population at an increased risk. However, the high rates of COVID-19 transmission coupled with the continual emergence of new variants (e.g., Delta and Omicron) presents unique challenges [5, 14] and requires multidisciplinary surveillance strategies (i.e., wastewater surveillance) and collaboration amongst partners.

Wastewater monitoring, a re-emerging method of surveillance, has gained significant traction during the COVID-19 pandemic. This type of surveillance has successfully provided an early warning sign of the potential spread of COVID-19 within university settings, allowing for rapid implementation of mitigation strategies [15–21]. While wastewater surveillance methods have differed slightly, it is well-understood that viral shedding can be detected in wastewater samples from both symptomatic and asymptomatic individuals. In several instances of this type of surveillance on university campuses, SARS-CoV-2 has been detected in wastewater collected around dormitories on average seven days before human cases were identified [16, 18].

The first cases of the Delta variant were identified in the US in July 2021 and Delta quickly became the dominant SARS-CoV-2 variant across the nation. Surveillance data showed that Delta was significantly more contagious, resulted in more severe infections, particularly in unvaccinated individuals, and was associated with breakthrough infections [22–24]. Robust genomic sequencing efforts in Arizona throughout the second year of the pandemic revealed the establishment and circulation of several Delta sub-lineages (e.g., B.1.617.2, AY.3, AY.25, AY.44, AY.103). One of these sub-lineages, AY.25, has mostly been seen in North America [25] and comprised approximately 7% of sequenced genomes in Arizona during fall 2021 (https://pathogen.tgen.org/covidseq-tracker/arizona_trends.html).

Routine genomic surveillance and ongoing phylogenetic analysis of sequence data from positive clinical samples identified an AY.25 clonal cluster associated with a university and the surrounding community throughout August to December 2021. The identification and characteristics of the outbreak are described here, with highlights on the integration of clinical, epidemiologic, genomic, and wastewater surveillance data. This report demonstrates the importance of implementing and maintaining one health surveillance strategies to strengthen response actions.

## Methods

### University infrastructure, testing and mitigation efforts

Beginning the week of August 15, 2021, a moderate sized university in Arizona opened its doors to the approximately 20,000 students for the Fall semester. The university hosted various

summer programs, including sports team trainings and an engagement and mentorship program for first-generation college students (mid-June through mid-July). Individuals participating in these summer programs were housed in dormitories throughout the campus. Although several students remained in the area throughout the summer in off-campus housing, the majority moved back to university dormitories August 19th through August 21st ("move-in weekend").

The university had implemented several prevention strategies in the previous year of the pandemic, including mask policies, social distancing, reduced dining hall capacity, hybrid learning, encouraged vaccination, and mitigation testing. The mitigation testing was an ongoing effort to identify asymptomatic cases among students, faculty, and staff, and involved a random selection of 2,500 individuals (both vaccinated and unvaccinated) each week to participate in voluntary COVID-19 screening. This self-collected saliva-based PCR testing was available 6 days a week at a designated on-campus location (through the end of the semester); testing for the general community was also available at this location through appointments. Lastly, campus health services provided rapid testing for symptomatic individuals.

Students living on and off campus were also encouraged to self-report positive SARS-CoV-2 test results to the university. Based on CDC guidelines at this time, students who tested positive for COVID-19 were required to isolate for 10 days and unvaccinated close contacts were quarantined for at least 8 days in designated isolation dormitories or off-campus (home of parent or friend). Mandatory dormitory-wide mitigation testing was triggered if more than 1% of the population residing in that dormitory tested positive. The campus health services team conducted extensive contact tracing to investigate close contacts and determine the need to implement additional and targeted interventions.

Contact tracing for COVID-19 positives cases in the community was also regularly conducted by the local public health officials. Detailed information was collected from the Arizona Medical Electronic Disease Surveillance and Intelligence System (MEDSIS) including demographics, university affiliation, on-campus residence if applicable, symptoms, travel history, participation in social gatherings or activities, close contacts, and vaccination status.

### SARS-CoV-2 phylogenetic analyses

Whole genome sequencing of SARS-CoV-2 PCR positive samples is routinely performed by several laboratories in Arizona [26] and by the Centers for Disease Control and Prevention, using similar methodologies [27]. SARS-CoV-2 genomes publicly available through GISAID (S1 Table) from these sequencing efforts were utilized to build phylogenetic trees using the SARS-CoV-2 specific workflows (https://github.com/nextstrain/ncov/) in Nextstrain [28]. Genomes with less than 70% breadth of coverage relative to the Wuhan reference genome (GISAID: EPI_ISL_402125, NCBI: NC_045512.2) were excluded; the G142D site, known to be associated with sequencing error, was also masked [23]. Maximum likelihood inference was executed with the IQ-TREE method (http://www.iqtree.org).

A regular review of the phylogenetic assignments identified what appeared to be a clonal cluster (i.e., multiple genomes with 0–2 single nucleotide polymorphisms (SNPs) between them being identified within the same geographical area) within the AY.25 sub-lineage of Delta. Genomic analyses for the distinct AY.25 outbreak sub-clade was shared with the local public health department. Epidemiologic, clinical, and exposure information routinely gathered through contact tracing and investigations was collated and matched with the available genomes. Duplicate genomes for the same SARS-CoV-2 positive case (e.g., repeated positive samples for same person that had been submitted for sequencing) were removed from the phylogeny. In these cases, genomes with the earliest sample collection dates were included.

The Office of Research Compliance & Quality Management (ORCQM) with the Translational Genomics Research Institute (TGen) reviewed this work and determined that it is not considered Human Subjects Research. Informed written or verbal consent was therefore not required, as this work was conducted in collaboration with a local public health agency as a public health surveillance activity, involving viral genomic sequencing and analysis of de-identified remnant specimens.

## Wastewater monitoring

Wastewater surveillance at the university began in July 2020, and during the period of August 13th through December 8th of 2021, samples were collected twice weekly from 16 sites strategically selected to assess SARS-CoV-2 levels in different residence halls. Using a modified Moore Swab style of approach [29], Tampax Pearl Ultra Absorbency tampons were suspended in wastewater flows outside of residence halls for 48–72 hours. Subsequently, tampons were pulled from the wastewater flow and placed into a contained filled with 20mL of deionized water which was shaken vigorously for two minutes. One mL of liquid was concentrated using the Zymo Research Water Concentrating Buffer, and extractions were carried out using the Quick-DNA/RNA Miniprep Plus Kit. Extracted material was evaluated in triplicate for SARS-CoV-2 using the CDC N1 RT-qPCR assay, and our standard quality control measures were applied, including embedded reagents blanks(n = 3), no-template controls (n = 3), and positive controls (IDT 2019-nCoV_N Positive Control plasmid, catalog number: 10006625, at dilutions of 1:10, 1:100, and 1:1000) each time we process samples. Samples with cycle threshold values less than 35 were subsequently sequenced to identify variants of concern, using the xGen™ SARS-CoV-2 Amplicon Panel (previously the Swift Normalase Amplicon SARSCoV-2 Panel) with unique dual indices from Integrated DNA Technologies. Resulting sequencing reads were aligned to the Wuhan reference genome (GISAID: EPI_ISL_402125, NCBI: NC_045512.2) using Bowtie2 [30] and mutations of interest were called using Sam Refiner [31].

## Results

### Identification and characteristics of AY.25 outbreak cases

From August 22nd to December 4th, 2021, a total of 1,299 COVID-19 cases were identified among individuals affiliated with the university and 1,577 cases were individuals tested from the general community. Approximately 70% of positive case samples from this testing were submitted for whole genomic sequencing. Retrospective phylogenetic analysis of these publicly available SARS-CoV-2 genomes combined with sequencing data from other Arizona laboratories revealed several SARS-CoV-2 Delta variant sub-lineages (assigned by Pangolin) circulating within the university setting and surrounding county, including B.1.617.2 (n = 851), AY.25 (n = 451), AY.44 (n = 339), and AY.103 (n = 295) during the same timeframe. Of the 451 sequenced genomes characterized as AY.25, there was an even distribution of this lineage among the community (n = 228, 50.5%) and university (n = 223, 49.5%).

Review of these Delta sub-lineages within the larger phylogeny (Fig 1) revealed a distinct monophyletic clade of the AY.25 sub-lineage, comprising 379 total genomes (S1 Table). Of these sequenced genomes, 49.8% (n = 189) were from SARS-CoV-2 positive cases associated with the university (case rate of 5,883 per 100K) and 42.2% (n = 160) were associated with the community (case rate of 6,349 per 100K); the affiliations of the remaining (n = 31) were unknown. Additionally, when comparing the number of positive cases associated with and reported by the university, 14.5% (189/1,299) were found to be a part of this AY.25 outbreak.

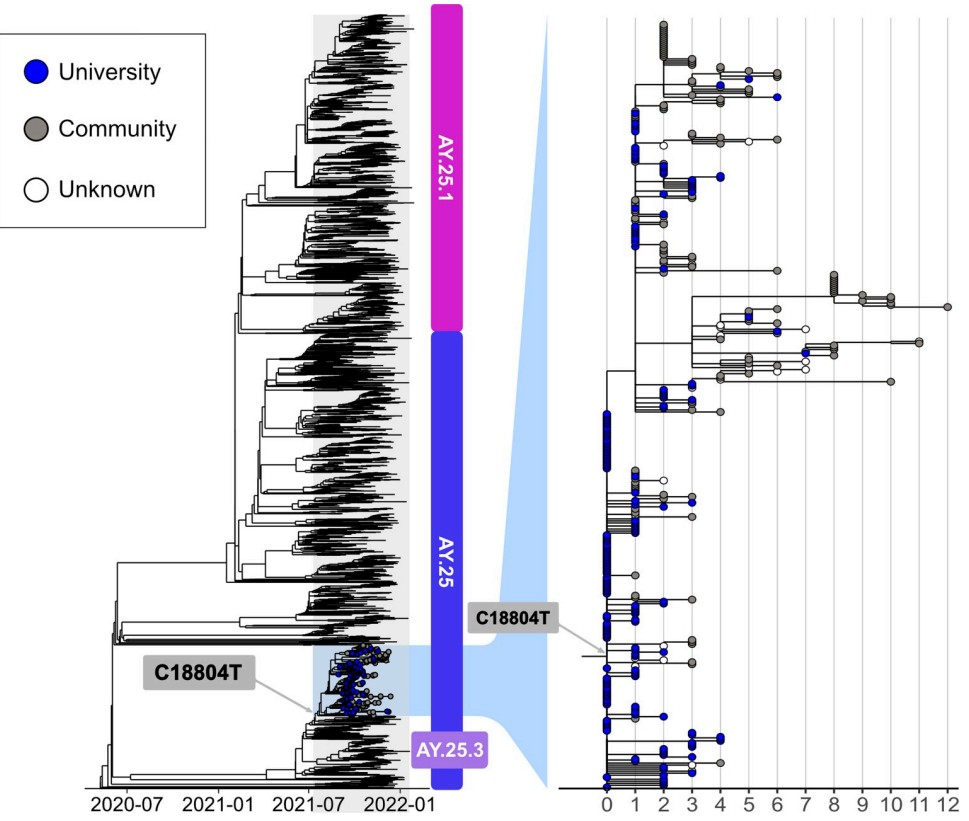

**Fig 1. Maximum likelihood phylogenetic tree of the AY.25 SARS-CoV-2 genomes (n = 379) associated with the university outbreak, highlighting the clade-defining mutation (C18804T).** The Wuhan-1 genome was used as a reference (EPI_ISL_402125, NCBI: NC_045512.2) and SARS-CoV-2 genomes (publicly available through GISAID) of the AY.25 lineage in Arizona are shown for context.

Among just the AY.25 university associated cases, 91.0%, (172/189) were under 25 years of age, while the ages of the community cases were more evenly distributed across age groups and ranged from 5–82 years. 51.9% (98/189) of the university cases were female and 62.4% were white, non-Hispanic (118/189). Table 1 further displays relevant demographic, clinical, vaccination, and exposure characteristics of university and community associated cases; however, it is important to note that complete epidemiologic data was not available for every case associated with this outbreak.

89.9% (170/189) of university associated cases occurred among first and second-year undergraduate students living on and off campus. On-campus students comprised 65.3% (111/170) of the AY.25 university cases, with the vast majority (81.9%, 91/111) residing in four freshman dormitories. These were co-ed traditional style campus dormitories with two students per room, and a community bathroom, laundry room, kitchen, and lounge. 59/170 students (34.7%) lived in off-campus housing within commuting distance from the university during the semester. Only three individuals (1.6%) associated with the university were employees.

The first SARS-CoV-2 positive case associated with the outbreak for which a viral genome was produced had a collection date of August 30th (7 days after the fall semester began on August 23, 2021). During an interview with public health, this case named at least four other individuals ("close group of friends") whom they had close contact with during their infectious period; two of which resided in the same dormitory (Dormitory A) and one of whom resided in Dormitory B. Contact tracing efforts further confirmed these linkages and identified what is

Table 1. Characteristics[+] of university and community cases associated with the AY.25 outbreak, Arizona, August—December 2021.

| Characteristic | University (189) | Community (160) |
| --- | --- | --- |
| | N (%) | N (%) |
| **Age (N = 266)** | | |
| ≤24 years | 172 (91.0) | 37 (23.1) |
| 25–44 years | 6 (3.2) | 37 (23.1) |
| ≥45 years | 1 (0.5) | 13 (8.1) |
| **Gender (N = 266)** | | |
| Male | 80 (42.3) | 55 (34.4) |
| Female | 98 (51.9) | 31 (19.4) |
| **Symptom Status (N = 158)** | | |
| Asymptomatic | 12 (6.3) | 3 (1.9) |
| Symptomatic | 103 (54.4) | 40 (25.0) |
| **Hospitalization Status (N = 168)** | | |
| Not Hospitalized | 116 (61.4) | 50 (31.3) |
| Hospitalized | 1 (0.5) | 1 (0.6) |
| **Vaccination Status** | | |
| Vaccine Received (N = 76) | 43 (22.8) | 33 (20.6) |
| ≥2 Doses Vaccine Received (N = 76) | 36 (19.0) | 24 (15.0) |
| Missing Vaccine Data (N = 190) | 136 (72.0) | 54 (33.8) |
| Vaccine Breakthrough (N = 67) | 39 (20.6) | 28 (17.5) |
| **Type of Residence/Congregate Setting (N = 170)** | | |
| On Campus (Dormitory Living) | 111 (58.7) | N/A |
| Off-Campus | 59 (31.2) | N/A |
| **Exposure History** | | |
| Close Contact w/ Positive Case (N = 158) | 57 (30.2) | 23 (14.4) |

[+] Proportions are summarized by affiliation. Demographic and exposure data was not available for every case associated with the outbreak. The "N" listed next to each characteristic is reflective of known data from public health interviews for both university and community associated cases.

believed to be the index case of this epidemiologic cluster (first case in Dormitory A and the university during this time)–a symptomatic student from out-of-state, whose SARS-CoV-2 positive sample collected on August 28th, but was not sequenced (Fig 2). A total of 10 cases associated with this outbreak were identified in Dormitories A and B throughout the Fall 2021 semester.

The AY.25 outbreak peaked in the university population during the week of September 19th– 25th (Calendar Week 38), and cases associated with the community peaked during the week of October 17th– 23rd (Calendar Week 42). Of the interviewed university associated cases, 36.1% (57/158) were aware they had close contact to a positive COVID-19 case. Various events and social gatherings within the first few weeks of the semester, including welcome week activities, sporting events, and house parties. While vaccination information was not available for all outbreak associated cases, 17.7% (67/379) vaccine-breakthrough cases were identified (Table 1).

## Phylogenetic analysis of AY.25 outbreak

Throughout the academic semester, this particular AY.25 clone, genomically defined by a single synonymous SNP, C18804T (Fig 1), became established and evolved, following its

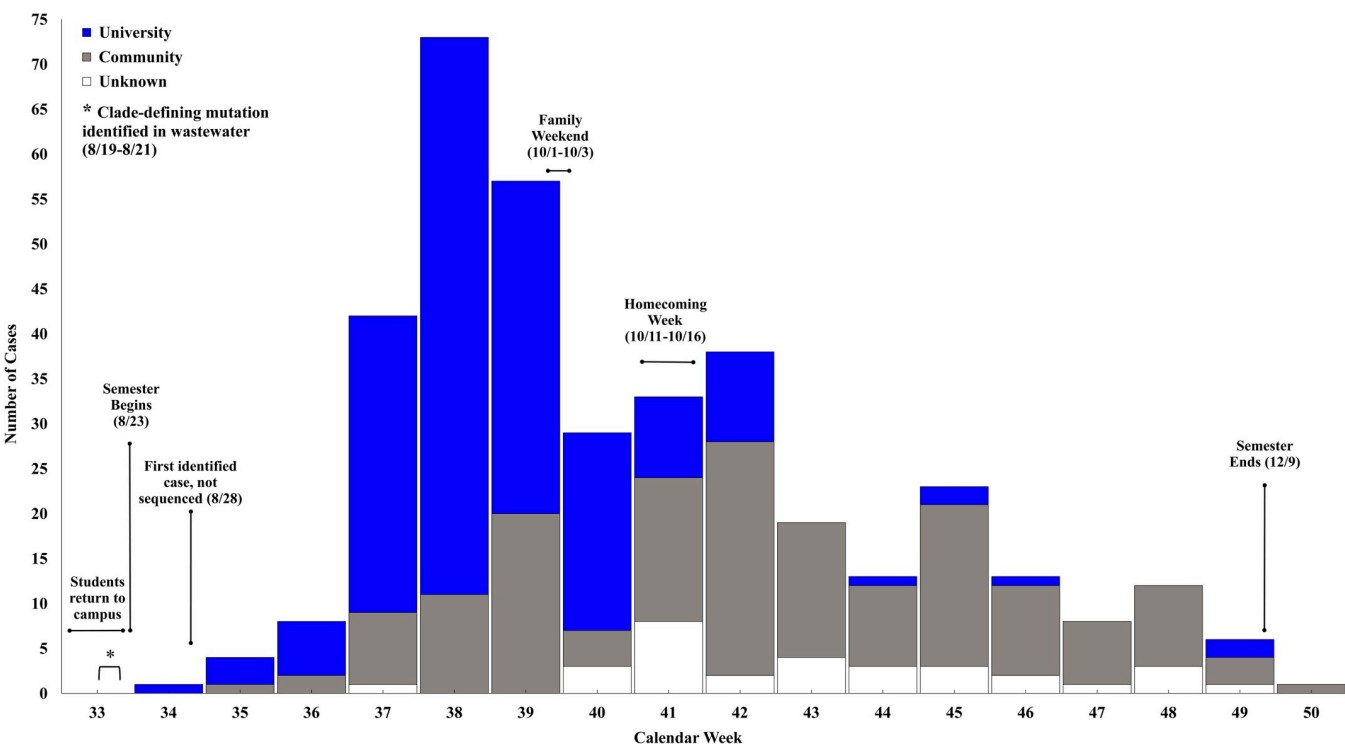

**Fig 2. COVID-19 cases (N = 380) associated with AY.25 outbreak by original specimen collection date, Arizona, August—December 2021.**

importation into the university setting. 95 genomes are identical, and the remaining AY.25 sequences in this phylogeny include up to 12 total SNP mutations across the 379 genomes identified during the outbreak from the ancestral genomes.

The maximum likelihood phylogeny indicates numerous instances of secondary spread occurred, however, transmission of this AY.25 clone was generally sporadic throughout the community and did not extensively impact overall COVID-19 trends in the region. One non-university secondary transmission chain contained 37 viral genomes and was defined by an amino acid in the Orf1a region. Twelve of these 37 genomes (nine were identical and three had one additional SNP) were associated with individuals attending a local elementary school (collection dates of SARS-CoV-2 positive samples ranged from October 13-20[th]).

## Wastewater surveillance confirms AY.25 outbreak clone

As expected, wastewater sampling conducted as a baseline the week prior to students moving into university dormitories yielded no SARS-CoV-2 positive campus sites. However, the AY.25 outbreak clone, was detected in two wastewater samples collected between August 19[th] through 21[st] (move-in weekend) (Fig 2). This sample was collected seven days prior to the university's index case sample being collected, and in two different residence halls (Dormitory C and D) than the initial human cases on campus were associated with. SARS-CoV-2 virus in these wastewater samples, and seven others collected between August 30th through September 23rd contained the clade-defining mutation (C18804T). The first human cases associated with Dormitories C and D had samples collected September 3[rd] and 14[th] (neither of which were sequenced), and a total of 34 cases associated with this outbreak were identified in these two dormitories throughout the Fall semester.

## Discussion

The introduction, establishment, and transmission of infectious diseases leading to outbreaks on university and college campuses are inevitable, given the intimate interactions of students in dormitories and involvement in various extracurricular activities. Therefore, the integration of complementary one health approaches (e.g., contact tracing, genomic epidemiology, and wastewater surveillance) is invaluable for early detection and response to potential threats in these settings.

This report documents a clonal outbreak of the AY.25 Delta sub-lineage impacting a local university and surrounding community identified through ongoing phylogenetic analysis, while the outbreak was occurring. Genomic surveillance and extensive contact tracing by public health and campus health services provided strong evidence that this outbreak clone was 1) amplified by extensive interactions among students and 2) not maintained solely within the university population. Genomic epidemiologic analyses confirmed several secondary transmission chains to individuals throughout the community. However, while this distinct AY.25 sub-lineage reported here was well-established (and public health data indicates the AY.25 parent lineage was the second most common lineage circulating in the area), it is important to note that this outbreak was not a driver of COVID-19 transmission in the community as a whole. Furthermore, this outbreak did not influence patterns of how the virus was circulating in the general community during the Fall 2021 timeframe. Although not detailed in this report, reciprocal transmission events of other SARS-CoV-2 lineages from the community to individuals affiliated with the university likely occurred given the nature of interactions among these two sub-populations.

Most of the university affiliated cases were first-year undergraduate students living on and off campus, but frequently interacting with each other in residence halls, at campus events and parties during the early weeks of the semester. Previous analyses [13] found that COVID-19 cases were not correlated with dormitory living and in-person classes, and that infections have largely been acquired in off-campus settings. While this may be representative of the nature of COVID-19 outbreaks amongst some universities, evaluation of this AY.25 outbreak showed that at minimum, the initial rapid growth phase of the outbreak occurred among students who had close interactions with each other in four dormitories (i.e., Dormitories A-D). Continued transmission of the virus was then through additional student interactions once the AY.25 clone was established in that population. Of interest, out of the 111 student cases living on-campus, 47.7% (n = 53) lived in the same dormitory. This could be a potential artifact of the mitigation testing, but also a result of the close contact students had with each other in that dormitory. The findings from this outbreak also demonstrate that there is limited risk of transmission to staff and university faculty, providing further evidence that viral spread in a classroom setting is not always a driver of university outbreaks.

Furthermore, unlike the report by Valesano, et al (2021) [32] which found that university cases in Michigan did not contribute enhanced transmission in the community, the viral clone discussed here was identified in more than 40% of cases with no known association with the university (i.e., community). This may in part be due to the relatively small size of the surrounding city (estimated population of ~75,000) and its tourism-based economy. Although low levels of ongoing transmission occurred throughout the remainder of the semester, mostly within the community, the AY.25 outbreak began to wane by the end of October 2021, likely in part to the enhanced efforts deployed by the university.

Data from wastewater surveillance efforts, which had been in place across the university for almost two years to provide an early warning for COVID-19 case clusters, adds an important layer to this outbreak. Following the phylogenetic analysis of clinical samples, genomic

characterization of positive wastewater samples collected from multiple dormitory sites across campus during move-in weekend and the subsequent weeks showed the presence of the clade defining mutation. These findings thereby confirmed the presence of the AY.25 outbreak clone in three different dormitories (including Dormitories C and D) prior to the initial identification of human cases. The community had experienced a steady level of COVID-19 cases throughout the summer, and limited activities occurred on campus, except for the previously mentioned sports-related trainings and mentorship programs.

While hundreds of Delta variant sub-lineages were characterized throughout 2021, AY.25 was not a dominant sub-lineage across the U.S.; however, in-depth phylogenetic analysis of outbreak cases associated with large public summer gatherings in Massachusetts revealed that AY.25 was responsible for a single cluster containing 84% of outbreak-associated genomes [33]. A second analysis of Delta variant associated vaccine breakthrough infections in New York identified a small cluster of AY.25 with a signature mutation in the spike protein, S112L [7]. According to Duerr et al (2022) [34], while S112L is not a Delta-defining mutation, it appears to be more commonly found in genomes associated with vaccine breakthrough cases. None of the viral sequences in the Arizona AY.25 university outbreak, including the 67 vaccine breakthrough cases, contained that genomic signature.

There are some important limitations to address. First, the university reported an adherence rate of 25–30% for weekly mitigation testing offered throughout the semester, and further noted that students (mainly those living off-campus) who acquired COVID-19 testing outside of campus health services may not have reported positive results, which could have impacted case identification, contact tracing and further prevention efforts. Second, complete epidemiologic information was not available for every positive case associated with this outbreak. Although an extensive effort was made to by public health and campus health officials to conduct thorough contact tracing, many cases are lost to follow-up. Lastly, the number of individuals associated with this outbreak is likely an underestimate given that not every infected patient was likely tested, and sequencing was not performed on every PCR-positive sample or performed on any antigen-based or rapid PCR positive samples. Despite these limitations, there is supporting evidence that the AY.25 clone was likely imported into the university setting just prior to the onset of the Fall semester, rapidly spread through a subset of the student population, and then subsequent spillover occurred in the surrounding community.

## Conclusions and public health implications

As we see promise of SARS-CoV-2 endemicity in the US, it is critical for universities to have guidance and policies in place for future outbreaks of COVID-19 and other communicable diseases. Implementation of multifaceted interventions has proven successful; these efforts included a combination of university wide screening programs, vaccination requirements (where applicable), social-behavioral policies (e.g., social distancing and masking), flexible education delivery models (e.g., hybrid learning options), and enhanced contact tracing to minimize spread before establishment in the campus community.

Additionally, collaboration and communication between universities and local public health entities is an important component of timely outbreak response. In the case described here, the university and public health department, having already worked closely throughout the pandemic, were able to easily identify and investigate clusters of cases within the dormitories. University testing data identified dormitories exceeding the 1% positivity threshold needed to require mitigation testing. This information was sent by the university to local public health workers who used case investigation data to verify that the potential outbreak met the necessary criteria for mandatory mitigation testing. Once verified, the university began

dormitory-wide mitigation testing in combination with the other established prevention strategies. Furthermore, increases in the presence of SARS-CoV-2 in wastewater samples resulted in the university enhancing their mitigation testing procedures around the impacted campus areas. The rapid spread of Omicron throughout universities [5, 35] highlights the continued need for these efforts as well as ongoing surveillance programs to identify early infections before symptomatic cases and widespread transmission occurs.

The incorporation of genomic epidemiology and wastewater surveillance can complement the more routine activities and advance the state of public health surveillance. Genomic surveillance and epidemiology provide the only empirical means for confirming the presence of a clonal outbreak and identifying which individuals are and, importantly, which are not part of an outbreak or cluster, especially, when multiple variants or pathogen strains are circulating. Analysis of the information gathered from cases that are known to be part of the same outbreak increases the ability to gain insight on how the virus was being spread among this population. Inclusion of non-traditional samples sources, such as wastewater, in genomic analyses can provide valuable early waring information or, at a minimum, provide additional context on likely outbreak sources and scope of spread. Wastewater-based epidemiology can also be applied to other infectious diseases of interest, such as influenza and enteric infections.

Understanding transmission patterns within the outbreak population helps identify improved disease prevention and mitigation strategies, increasing community resilience for future outbreaks. The potential threat of other infectious disease outbreaks on university campuses and in other settings presents an opportunity for future comprehensive one health data driven, targeted interventions.

## Supporting information

**S1 Table. GISAID Accession ID numbers for the AY.25 SARS-CoV-2 genomes (n = 379) associated with the university outbreak used to build the maximum-likelihood phylogenetic tree.**
(PDF)

## Acknowledgments

The authors wish to acknowledge the public health efforts of the local county health department and university campus health services during this pandemic to ensure the safety and well-being of the university population and community. The authors extend full gratitude to the executive team and administrative staff at the local university for their support of this work, review of the manuscript and collegial collaborations. We also would like to thank Taylor Martins from the Arizona Department of Health Services for his assistance in acquiring epidemiological data for cases associated with this outbreak. Lastly, we are extremely grateful to the GISAID Initiative and all its data contributors, specifically those from Arizona State University, Centers for Disease Control and Prevention, and Translational Genomics Research Institute laboratories responsible for obtaining SARS-CoV-2 positive specimens and generating viral genetic sequence data.

## Author Contributions

**Conceptualization:** Hayley D. Yaglom, Matthew Maurer, David M. Engelthaler.

**Data curation:** Jacob Hojnacki, Juan Monroy-Nieto, Samuel Packard, Daryn E. Erickson, Zachary A. Barrand, Kyle M. Simmons, Breezy N. Brock, Efrem S. Lim, Crystal M. Hepp.

**Formal analysis:** Hayley D. Yaglom, Jacob Hojnacki, Juan Monroy-Nieto, Daryn E. Erickson, Zachary A. Barrand, Kyle M. Simmons, Breezy N. Brock, Crystal M. Hepp.

**Investigation:** Hayley D. Yaglom, Matthew Maurer, Brooke Collins, Sandra Smith.

**Methodology:** Hayley D. Yaglom, Crystal M. Hepp.

**Project administration:** Hayley D. Yaglom.

**Resources:** Hayley D. Yaglom, Matthew Maurer, Brooke Collins, Jolene R. Bowers, Samuel Packard, Efrem S. Lim, Sandra Smith, Crystal M. Hepp, David M. Engelthaler.

**Supervision:** David M. Engelthaler.

**Writing – original draft:** Hayley D. Yaglom.

**Writing – review & editing:** Hayley D. Yaglom, Matthew Maurer, Brooke Collins, Jolene R. Bowers, Crystal M. Hepp, David M. Engelthaler.

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
