## [Decision Letter · Decision Letter 0]

16 Aug 2022

PONE-D-22-21082One Health Genomic Surveillance and Response to a University-Based Outbreak of the SARS-CoV-2 Delta AY.25 Lineage, Arizona, 2021PLOS ONE

Dear Dr. Yaglom,

Thank you for submitting your manuscript to PLOS ONE. After careful consideration, we feel that it has merit but does not fully meet PLOS ONE’s publication criteria as it currently stands. Therefore, we invite you to submit a revised version of the manuscript that addresses the points raised during the review process.

We look forward to receiving your revised manuscript.

Kind regards,

Ruslan Kalendar

Academic Editor

PLOS ONE

Journal Requirements:

1. Please ensure that your manuscript meets 

PLOS ONE's style requirements, including those for file naming. The PLOS ONE style templates can be found at 

"The NARBHA Institute and the Arizona Department of Health Services (AZDHS) provided funding support for this work. "

"We also wish to thank the NARBHA Institute and the Arizona Department of Health Services (AZDHS) for funding support for this work."

"The NARBHA Institute and the Arizona Department of Health Services (AZDHS) provided funding support for this work. "

Reviewers' comments:

Reviewer's Responses to Questions

**Comments to the Author**

1. Is the manuscript technically sound, and do the data support the conclusions?

Reviewer #1: Yes

2. Has the statistical analysis been performed appropriately and rigorously? 

Reviewer #1: No

3. Have the authors made all data underlying the findings in their manuscript fully available?

Reviewer #1: Yes

4. Is the manuscript presented in an intelligible fashion and written in standard English?

Reviewer #1: Yes

5. Review Comments to the Author

Reviewer #1: The authors of the manuscript PONE-D-22-21082 have conducted an interesting study. Some important points that need to be incorporated in the manuscript are mentioned below;

Please cut short the abstract mainly the background information. The background information could be limited to 2-3 lines followed by the aim of the study and the results obtained.

Is it possible to insert a graphical abstract to grab the attention of the readers/researchers?

Was there any method blank, extraction blank?

What about positive and negative controls?

What are the QA/QC measures taken into consideration?

Was the LoD determined?

Was any inhibition noticed?

Were there any replicates for the analysis?

Please incorporate statistical analysis.

Conclusion section is missing

Please increase the font size in the figures.

6. PLOS authors have the option to publish the peer review history of their article (what does this mean?). If published, this will include your full peer review and any attached files.

Reviewer #1: No

---

## [Author Response · Author response to Decision Letter 0]

31 Aug 2022

Please see the attached "Response to Reviewers" document. Thank you.

---

## [Decision Letter · Decision Letter 1]

5 Sep 2022

One Health Genomic Surveillance and Response to a University-Based Outbreak of the SARS-CoV-2 Delta AY.25 Lineage, Arizona, 2021

PONE-D-22-21082R1

Dear Dr. Yaglom,

We’re pleased to inform you that your manuscript has been judged scientifically suitable for publication and will be formally accepted for publication once it meets all outstanding technical requirements.

Kind regards,

Ruslan Kalendar

Academic Editor

PLOS ONE

---

## [Editor Report · Acceptance letter]

21 Sep 2022

PONE-D-22-21082R1 

One Health Genomic Surveillance and Response to a University-Based Outbreak of the SARS-CoV-2 Delta AY.25 Lineage, Arizona, 2021 

Dear Dr. Yaglom:

I'm pleased to inform you that your manuscript has been deemed suitable for publication in PLOS ONE. Congratulations! Your manuscript is now with our production department. 

Kind regards, 

on behalf of

Professor Ruslan Kalendar 

Academic Editor

PLOS ONE